# Relative importance of informational items in participant information leaflets for trials: a Q-methodology approach

Karen Innes, Seonaidh Cotton, Marion K Campbell, Jim Elliott, Katie Gillies

Health Services Research Unit, Institute of Applied Health Sciences, School of Medicine, Medical Sciences and Nutrition, University of Aberdeen, Aberdeen, UK

**Correspondence to**
Dr Katie Gillies;
k.gillies@abdn.ac.uk

## ABSTRACT

**Objectives** To identify which information items potential participants and research nurses rank as the most important, and the reasons for this, when considering participation in a randomised controlled trial.

**Design** Q-methodology approach alongside a think-aloud process. Using a vignette outlining a hypothetical trial, participants were asked to rank statements about informational items usually included in a participant information leaflet (PIL) on a Q-grid, while undertaking a real-time think-aloud process to elicit the underpinning decision processes. Analysis of quantitative data was conducted using descriptive statistics and qualitative data was coded using content analysis.

**Participants** 20 participants (10 potential trial participants and 10 research nurses).

**Setting** UK-based participants.

**Results** Ten research nurses and 10 potential trial participants provided data for the study. Both stakeholder groups ranked similar statements in their top three most important statements, with 'What are the possible disadvantages and risks of taking part?' featuring in both. However, considerable variability existed between the groups with regard to their ranking of statements of least importance. Participants identified that sufficient information to make a decision was secured using around 14 items. Participants also identified other items of importance not routinely included in PILs.

**Conclusions** This study has provided a unique insight into how and why different trial stakeholder groups rank informational items currently contained within PILs. These results have implications for those developing future PILs and those who develop guidance on their content; PILs should focus most on the information items that potential trial participants want and need to make an informed choice about trial participation.

## BACKGROUND

Research is an important part of the development of medicine, including the development of new treatments, services and technologies. In particular, randomised controlled trials (RCTs) are considered the gold standard for evaluating the efficacy and safety of new treatments and effectiveness of existing interventions.[1] [2] Central to the successful delivery of RCTs are the participants who agree to take

### Strengths and limitations of this study

► This study is one of the first to provide evidence on the importance of informational items prescribed in the regulatory guidance with regard to making an informed choice about randomised controlled trial participation to potential trial participants and research nurses.

► Our study used a novel methodology (Q methodology) to obtain rankings of informational items for participant information leaflets from different trial stakeholder groups, namely potential trial participants and research nurses.

► The solely UK-based self-selecting sample may hold different views to those in other countries with different social norms and cultures.

part. Strict regulations and legislation are in place governing the process of approaching and consenting potential participants to take part in order to ensure that their rights and interests are protected.[3] [4]

Seeking informed consent (usually prospectively) from potential participants is a prerequisite for their inclusion within almost all RCTs. A printed participant information leaflet (PIL) is a key document that aims to support the informed consent process. A PIL should provide the reader with clear and easy-to-understand information.[3] [4] Regulatory bodies have provided guidance on the inclusion of content which they deem to be required to ensure that the consent given is 'informed'.[3] [4] In addition to providing information about the proposed research, a PIL often provides a mechanism to support conversations about the trial between the potential participant and the researcher and/or health professional, allowing the participant the opportunity to ask any questions important to their decision and discuss the research in more detail.[5] However, the recruitment and consent process for some trials is such that a conversation between a researcher and potential participants is less likely (eg, postal or online recruitment) and

here the written information may have more influence. Ideally, the aim of the PIL should be to provide information to assist the participant in making a decision as to whether to take part in a trial or not.[5]

In the UK, current guidelines for PILs are set out by the Health Research Authority (HRA), the body established to ensure that the interests of patients who take part in research are protected and also to promote good quality research in the UK. The HRA's guidance lists 36 topic areas for suggested inclusion in PILs for research.[5] These 36 items were informed by legislation on informed consent for research and cover aspects such as: the purpose of the research; potential benefits and risks; the right to refuse or withdraw; and treatment alternatives.[3 4]

At present there is a lack of evidence about whether the topic areas identified in the HRA guidelines are perceived as important, or useful for decision making, from the participants' perspective. A systematic review by Kirkby et al[6] emphasised the lack of empirical evidence to support the items included in the HRA guidance with regard to what topics participants want to know about when considering taking part in research (not just trials). Furthermore Armstrong et al[7] suggest that PILs are written with the primary focus being regulatory review as opposed to a principal role in supporting participants' decision making.

Existing research also suggests that PILs may not be fit for purpose and that trial participants have a lack of understanding about key aspects of the trial.[8 9] This includes those participants who have consented and been recruited to trials and those who are considering participating in trials.[10] To date, existing research on PILs for trials has tended to focus on structure: redesigning and rewriting to improve readability and understanding and exploring easy-to-read consent statements versus standard consent statements or short versus long PILs.[8–10] The majority of this existing research has not questioned the information content (specified by the regulatory guidelines) that should be contained in PILs from the perspectives of potential participants and/or other stakeholders engaged in the trial consent process.

Aside from the participants themselves, research nurses (RNs) play a vital role in clinical trial delivery (certainly in the UK), particularly during the informed consent process. The role of an RN is that of the patient advocate, supporting any potential research participant throughout the research process. As RNs are routinely involved in seeking informed consent from potential research participants, they also have a unique insight into the topic areas and questions that may arise during the informed consent conversation. However, whether the informational items RNs perceive as being important to support decision making when discussing trials aligns with desires of potential participants is not known. Understanding whether these groups are similar or differ in their perspectives could provide important insights to improve the informed consent process for RCTs.

The aim of this study is to identify and assess which of the prescribed information items potential participants and RNs rank as the most important, and the reasons for this, when considering participation in a phase III RCT. A related objective was to explore whether there were any differences in how the information is ranked between the different groups.

## METHODS

This research study used a Q-methodology approach to determine the relative importance of informational items presented in PILs to potential trial participants (PTPs) during the informed consent process. Q-methodology uses a mixed methods approach that aims to identify shared views, opinions, beliefs and attitudes across a population, forcing people to trade off different dimensions and rank items in order of importance.[11] The Q-sort technique provides participants with a question/topic of interest and a set of associated relevant statements linked to the topic (the Q-set), which are then ranked by the participant according to what they feel are most and least important from their perspective in relation to the question posed by the researcher. The participant places statements onto a specialised grid (known as a response grid) and is asked to provide justification for placement through a 'think-aloud' process. Here, participants verbalise in real-time the thought processes underlying their choice of where to place each statement on the response grid.

In full Q-methodology, one is usually concerned with trying to identify how viewpoints cluster together; this is usually undertaken through the use of formal statistical Q-factor analysis.[11] In this study, however, we were more interested in the differences/similarities within and across the two stakeholder groups and the reasons why, so we did not proceed with the full factor analysis stage. We rather used descriptive statistics to summarise the perceived importance of items within stakeholder groups and further between stakeholder groups. As we did not use the full Q-methodology, we have described our study as using a Q-methodology approach.

### Scope of study

A vignette (see online supplementary file 1) was developed, which described a hypothetical phase III RCT of two treatments for a chronic condition, to help participants contextualise the Q-sort statements and enable them to provide their subjective opinions and points of view. Two vignettes were prepared (based on the same trial example but framed to the perspectives of the two stakeholder groups). The PTP group were asked to consider '*What information would be important to you when making a decision to take part?*' The RN group were asked, '*What information would be important to potential participants when making the decision to take part?*'

### Development of the Q-set

The Q-set of statements were developed using three sources of information: (1) the HRA guidance on

'Consent and Participation Information Sheets'[5]; (2) a published systematic review that identified empirical evidence to support what potential research participants want to know about research when considering participation[6]; and (3) a published scoping exercise that had identified desirable features for a centralised public information resource about clinical trials.[12] To avoid duplication of concepts, the development of the Q-set statements started with a mapping exercise where the individual informational items identified by Kirkby et al[6] and Langston et al[12] were mapped onto the list specified in the HRA guidance.[5] Given the generic focus of our vignette, a number of the more specialised HRA items (those which cover the particular circumstances of: radiation, pregnancy and breast feeding, young people and pregnancy, genetic research, screening and exclusion, adults not able to consent for themselves and commercial exploitation) were excluded from consideration. This resulted in a final total of 32 statements, which formed the Q-set.

A list of scripted prompts (related to each statement) were also developed to ensure consistency in response where further information or clarification was required by participants regarding what was meant by a particular statement allowing explanations to be standardised across interviews.

A 32-element Q-grid was then developed following a quasinormal distribution as per Q-methodology standards (see online supplementary file 2). The grid was split into three areas: columns 1–3 of the Q-grid represent the 'more important' items; columns 4–6 of the Q-grid represent 'neutral' items; and columns 7–9 of the Q-grid the 'less important' items. Statements were given a reference number and laminated. Three pilot Q-sorts and interviews were conducted to ensure comprehensiveness of the statements and prompts and ensure no overlap or duplication between statements.

## Sample size

For the purpose of this project, a sample size of 20 participants, 10 from each trial stakeholder group, was deemed appropriate. Typically, Q methodology uses relatively small samples of participants and the literature suggests that a 2:1 ratio of statements to participants (irrespective of stakeholder group) is favoured as a minimum. For example, a study with 40 statements would have 20 participants as a minimum. As this study has 32 statements, following the principle above, we would require an overall sample of approximately 16 participants in total as a minimum.[11]

## Participants

### Potential trial participants

PTPs were identified from the SHARE register. SHARE is a register of people who have an interest in taking part in research, developed by National Health Service (NHS) Research Scotland.[13] For the purposes of this project, people who lived within the NHS Grampian (NHSG) area

(the health board area of the lead researcher to allow face-to-face Q-sorts to be undertaken) were identified and invited in line with the current SHARE application process. The details of 17 potentially interested participants were provided to the research team by SHARE. All 17 potential participants were contacted by the researcher by telephone to arrange a convenient time for a Q-sort interview. Following this conversation, 10 participants expressed interest (seven declined further information) and were sent postal confirmation of the appointment time and a PIL for the Q-methodology study (available from the researchers on request). At the Q-sort interview, participants were provided with an opportunity to discuss the research and have any questions answered before completing a consent form and taking part in the card sort interview. All participants included in the study provided written consent.

### Research nurses

RNs were sought from the NHSG RN pool. Study information was provided to the NHSG RN manager who disseminated an invitation and the PIL relating to the study to the NHSG RNs email distribution list (n=100). Details of 12 interested nurses were received. Interested participants were asked to contact the researcher by email or telephone to arrange an appointment for a Q-sort interview. Following this, participants were sent an email with confirmation of the appointment time. At the Q-sort interview, RNs were provided with an opportunity to discuss the research project and have any questions answered before completing a study consent form and taking part in the Q-sort interview. All provided written consent.

## Data collection

One author (KI) conducted the Q-sort interviews between August 2015 and March 2016. All interviews were face-to-face and conducted at the University of Aberdeen. Q-sort interviews were audio recorded. At the start of the interview participants were presented with the trial vignette and the 32 statements (in random order each time) and asked to sort the statements into three initial piles: (1) those that they thought were important when considering whether to take part in the hypothetical phase III RCT; (2) those which they thought were less important; and (3) those which they had a neutral view about. Once the cards had been sorted into three piles, the participant was shown the Q-grid, given an explanation of how to place the cards onto the grid and asked to start placing them (ie, ranking in order of priority) while at the same time providing verbal explanation ('think aloud') as to why they were placing statements in a particular square of the grid. If participants were unsure of the meaning of any of the statements in the Q-set, the researcher used standardised prompts, described earlier, to aid understanding. On completion of the grid, the PTP group were asked if they felt any information was missing from the statements and also to indicate at which point on the grid

they would be able to make a decision about participation in the hypothetical RCT.

At the end of the task, participants were asked to complete a demographic details form and thanked for their participation. A photograph was taken of the completed response grid and a paper copy of the response grid completed by the researcher. Audio files were transcribed verbatim and anonymised accordingly.

## Data analysis

### Descriptive statistics

Data were collated across individual participants within each stakeholder group and used to calculate the following for each of the 32 items: (1) the median importance score (ie, the median position given by participants for that statement which could range from 1 to 9 (the higher the median importance score the less important the statement is that is, 1 most important, 9 least important); (2) the IQR around the median importance score; and (3) the range of scores for each item by group. These summary statistics allowed the statements to be ordered from most to least important for each of the trial stakeholder groups. The overall ranking of the statements was based on the median value; however, in the case where the median value was the same for more than one statement, the IQR was considered (and if necessary the range) in order to determine order. Differing views on individual items between the PTP and RN group were defined as 'discordant' if they exhibited a difference in the median rankings of ≥2 points between the groups. The PTP group were asked how many information cards they would require to make a decision about trial participation. These data were collated, and medians and a range were calculated.

### Qualitative analysis

Transcripts were read and reread to ensure complete familiarity with the transcripts. Text within the transcripts was coded by Q-set statement number using a content analysis approach.[14] Quotes were selected that illustrated reasons for ranking for the overall group majority or any outliers. Transcripts from the RNs and potential participants were initially considered separately but were then systematically compared for areas of agreement or disagreement.

## Patient involvement

Patients were not involved as research partners in the design, data collection or data analysis phases of this research. A patient research partner (JE) was involved in the drafting of the manuscript for publication. Participants in the research will be offered a summary of the results of the study.

## Approvals

All interview participants provided their signed consent, which included consent for anonymised quotes from their interviews to be published.

## RESULTS

### Participant characteristics: PTPs

Seventeen PTPs were approached through the SHARE database, and 10 consented to take part in this research project. The 10 PTPs had a mean age of 49.4 years (range 34–73 years). Five men and five women were interviewed; men had a mean age of 59.2 years, and women had a mean age of 39.6 years. Education levels varied between this group—four participants had secondary education (eg, O level, General Certificate of Secondary Education (GCSE) and Highers); one of these four had also completed an apprenticeship. The remaining six had completed higher education (eg, a degree). Seven PTPs had no previous experience of research. Q-sort interviews took an average of 38.7 min (range 23.6–62.3 min).

### Participant characteristics: RNs

One hundred NHSG RNs were invited through the RN manager email distribution list, and 12 consented and took part in this research project. Data from 10 of the 12 RNs is presented in the analysis due to an early change in the study documentation affecting the data from two of the participants. The 10 RNs whose data were included in the analysis were all female and had a mean age of 40.4 years (range 28–59 years). All had at least higher education (eg, a degree) and the range of research they had worked on varied from observational studies to Clinical Trial on an Investigational Medicinal Products (CTIMPs). Q-sort interviews took an average of 42.2 min (range 24.1–62.2 min). Summary characteristics of study participants are presented in table 1.

### Ranking of statements

Overall ranking summaries are presented for the potential participant group (table 2) and RN group (table 3).

| Table 1 | Summary participant characteristics | | |
|---|---|---|---|
| | **Potential trial participants** | | **Research nurses** |
| Age (median; range | 49.4 years (34–73 years) | | 40.4 years (28–59 years) |
| Gender (% female) | 5 (50) | | 10 (100) |
| Education (%) | Secondary | 30 | Secondary |
| | Apprenticeship | 10 | Apprenticeship |
| | Higher | 60 | Higher 100 |
| Involvement in research | Three previously participated in research | | CTIMPS Interventional non-CTIMPS Observational |
| Q-sort interview (median min:sec) | 38.7 min (range 23.6–62.3 min) | | 42.2 min (range 24.1–62.2 min) |

CTIMPs, Clinical Trial of an Investigational Medicinal Products.

**Table 2** Potential trial participants: ranking of statements (from most to least important)

| Statement | Rank | Median | IQR | Range |
|---|---|---|---|---|
| What are the possible side effects of trial treatment? | 1 (most important) | 2 | 1.5–3.5 | 1–5 |
| What are the possible disadvantages and risks of taking part? | 2 | 2 | 2–3 | 1–4 |
| What will I have to do? | 3 | 2.5 | 2–4 | 2–5 |
| What are the possible advantages of taking part? | 4 | 3 | 2–4 | 2–5 |
| What is the treatment that is being tested? | 5 | 3 | 2–4 | 1–7 |
| What will happen to my treatment when the research study stops? | 6 | 3 | 2.5–4 | 2–4 |
| How will my treatment be decided? | 7 | 3.5 | 3–5.5 | 2–7 |
| What will happen to me if I take part? | 8 | 4 | 1–5.5 | 1–7 |
| What is the purpose of this study? | 9 | 4 | 2–4 | 1–7 |
| Will I know what treatment I am on? | 10 | 4 | 3–7.5 | 3–9 |
| Has the scientific quality of study been checked? | 11 | 4.5 | 3–5.5 | 2–8 |
| What are the alternatives for treatment? | 12 | 4.5 | 3–6 | 3–7 |
| What happens if relevant new information becomes available? | 13 | 5 | 3–6 | 1–7 |
| Will my general practitioner be told? | 14 | 5 | 4–6.5 | 4–4 |
| What will happen to the results of the study? | 15 | 5 | 4–6.5 | 3–7 |
| Who has overall responsibility for the study? | 16 | 5 | 4.5–5 | 4–7 |
| Who has approved the study? | 17 | 5 | 5–6 | 2–7 |
| Do I have to take part? | 18 | 5.5 | 3.5–8 | 2–9 |
| Who could I contact for further information? | 19 | 5.5 | 4–6 | 4–7 |
| Who will have access to my data? | 20 | 5.5 | 4.5–7 | 3–7 |
| What if I have a complaint? | 21 | 5.5 | 5–7.5 | 4–9 |
| Why have I been invited? | 22 | 6 | 3.5–7.5 | 2–8 |
| Will my taking part in the study be kept confidential? | 23 | 6 | 4.5–7 | 3–8 |
| Will information from my existing medical records be accessed? | 24 | 6 | 4.5–7 | 2–8 |
| What will happen if I don't want to carry on with the study? | 25 | 6 | 5–6.5 | 4–7 |
| How have patients and the public been involved in the design of the study? | 26 | 6 | 5–7 | 4–7 |
| How will data be stored and disposed of? | 27 | 6 | 5.5–7 | 4–8 |
| What is involved in the consent process? | 28 | 7 | 5–8 | 4–9 |
| Who is funding the research? | 29 | 7 | 5.5–8 | 3–9 |
| Will expenses be reimbursed? | 30 | 8 | 5.5–8 | 5–8 |
| Will there be any impact on any insurance policies? | 31 | 8 | 5.5–8.5 | 3–9 |
| Will I receive any payments for taking part? | 32 (least important) | 8 | 6.5–8.5 | 6–9 |

### Top-ranking items: the most important information

There were several similarities between the RN and PTP groups in terms of the statements that they ranked as most important. PTPs ranked 'What are the possible side effects of trial treatment?' as their most important item, with RNs ranking it as fourth. Some of the reasons cited by PTPs for this being the most important related to their own personal safety, not being hurt and knowing the types of events they should report to the trial team

> … if it was going to be taking medication or if it was going to be some other sort of new treatment, it would be important to know as much as you could about what possibly might go wrong with it, so that you can protect yourself. (PTP20 – ranked in column 1)

RNs also reported trial participants want to know about side effects but that, in their perspective, this only mattered to a small number they ranked it lower.

> There has been a very few handful who have asked me for some data of how many percent have had side-effects or how many in the overall study how many – I have had questions but it's just that it's such a small rare quantity of people. (RN5 – ranked in column 4)

**Table 3** Research nurses: ranking of statements (from most to least important)

| Statement | Rank | Median | IQR | Range |
|---|---|---|---|---|
| What are the possible disadvantages and risks of taking part? | 1 (most important) | 2 | 2–4 | 2–4 |
| What is the purpose of this study? | 2 | 2 | 2.5–4 | 1–6 |
| What are the possible advantages of taking part? | 3 | 2.5 | 2–3.5 | 1–4 |
| What are the possible side effects of trial treatment? | 4 | 2.5 | 2–4 | 1–7 |
| What is the treatment that is being tested? | 5 | 3 | 1.5–4 | 1–4 |
| What will I have to do? | 6 | 3 | 2.5–4 | 2–4 |
| Do I have to take part? | 7 | 3 | 2.5–4.5 | 2–6 |
| What will happen to me if I take part? | 8 | 3 | 3–3.5 | 1–4 |
| How will my treatment be decided? | 9 | 3 | 3–4.5 | 2–5 |
| Why have I been invited? | 10 | 3.5 | 1–4 | 1–7 |
| What are the alternatives for treatment? | 11 | 4 | 3–4 | 2–5 |
| Will I know what treatment I am on? | 12 | 4 | 3–5 | 2–5 |
| What will happen to my treatment when the research study stops? | 13 | 4.5 | 4–5 | 3–5 |
| What happens if relevant new information becomes available? | 14 | 5 | 4–6.5 | 3–7 |
| What will happen if I don't want to carry on with the study? | 15 | 5 | 4.5–5 | 3–7 |
| Will information from my existing medical records be accessed? | 16 | 5 | 5–6 | 5–8 |
| Will there be any impact on any insurance policies? | 17 | 5 | 5–6 | 4–7 |
| Will expenses be reimbursed? | 18 | 5 | 5–6.5 | 4–7 |
| Will my taking part in the study be kept confidential? | 19 | 5.5 | 4–6 | 4–7 |
| Will my general practitioner be told? | 20 | 5.5 | 4–6.5 | 3–7 |
| What is involved in the consent process? | 21 | 6 | 4.5–6 | 2–7 |
| Who will have access to my data? | 22 | 6 | 5–6.5 | 5–7 |
| Will I receive any payments for taking part? | 23 | 6 | 5–6.5 | 5–8 |
| Who could I contact for further information? | 24 | 6 | 5–7 | 4–7 |
| What will happen to the results of the study? | 25 | 6 | 5.5–7 | 5–7 |
| What if I have a complaint? | 26 | 6.5 | 5.5–7 | 5–7 |
| Who has overall responsibility for the study? | 27 | 7 | 5.5–7 | 5–8 |
| How will data be stored and disposed of? | 28 | 7 | 5.5–8 | 5–8 |
| Who is funding the research? | 29 | 8 | 7.5–9 | 7–9 |
| Has the scientific quality of study been checked? | 30 | 8 | 8–8.5 | 7–9 |
| How have patients and the public been involved in the design of the study? | 31 | 8 | 8–8.5 | 6–9 |
| Who has approved the study? | 32 (least important) | 8 | 8–9 | 7–9 |

With regard to the second most important item, PTPs ranked 'What are the possible disadvantages and risks of taking part?' with RNs ranking it in first place. Although the position of the ranking is different between the groups the reasons provided were similar and related to benefits for self, while weighing up any potential negative consequences.

Well, I think I'd have to hear them both and then decide, you know? So, say, for example, you said with the advantages, it could improve your condition and the disadvantages were… you might get headaches with it or something, so it depends on the strengths of both. (PTP18 – ranked in column 2)

I think it's kind of almost maybe a sort of selfish kind of individual kind of thought of what does this mean for me rather than looking at the bigger picture of what the study is actually about. (RN1 – ranked in column 2)

PTPs ranked 'What will I have to do?' as the third most important statement highlighting the importance of knowing what would be expected of them, whereas RNs ranked this item in position six but with similar reasoning regarding expectations.

… just to make sure it wasn't going to involve too much from what would be the normal sort of scenario,

make sure that I wasn't committing to something that maybe… on top of something that might already be quite stressful or is going to add a lot of work or time… (PTP7 – ranked in column 3)

… with a chronic condition that patient's not that concerned about the end point of the study, just about getting an option for treatment. So I think they would actually want to know 'what will I have to come in and contribute, how much work will it be?' (RN7 – ranked in column 4)

The second and third most important items ranked by RNs did not feature in the PTPs top three. RNs ranked 'What is the purpose of this study?' in position number 2, stating the importance of highlighting to potential participants how the trial is relevant to them. However, PTPs ranked this statement in position number 9; the rationale being this statement has less to do with them as individuals. These items exhibited the biggest difference between groups in terms of items in each group's top 3, which is not surprising when considering the individual groups interpretations.

I feel this is the most important to let the patients know what we are trying to do, what's the purpose of doing the study to begin with. A bit of explanation as to why we're doing it in the first place. (RN3 – ranked in column 1)

'What the purpose is?' probably just to know whether it was something they were going to continue doing, or if it was just a trial and a kind of… guinea pig situation, just to see what happened. I suppose, knowing that if you could help other people with a similar condition, it might sort of give you the incentive to help or be part of it. (PTP17 – ranked in column 4)

In third place, RNs ranked 'What are the possible advantages of taking part?' as important, while PTPs ranked this statement as their fourth most important statement. Although in slightly different overall position, both RN and PTP gave similar reasons for their ranking, linked to balancing and weighing up of consequences.

So it may be that this drug won't be available to them, it's not going to be available to them if they don't take part so it's important that they know that, that there may be an advantage in the sense that they won't have access to this drug. (RN4 – ranked in column 4)

I would want to know the worst case scenario and then I'd probably ask after that what would be the benefits, because I would assume that there were going to be benefits, I guess. (PTP7 – ranked in column 3)

### Lowest ranking items: the least important information

PTPs ranked 'Will I receive any payments for taking part?' as the least important statement in position 32 with reasoning related to expectations of volunteering not requiring payment and opportunities for treatment outweighing remuneration. In comparison, RNs ranked

this in position 23 with some highlighting this as a potential incentive for patients to participate or provide outcome data.

Well, I volunteered so I don't expect to get paid for volunteering to do something. That's why I say that's the least important. (PTP13 – ranked in column 9)

I don't think patients are also that concerned about being reimbursed for taking part in the study. I think the benefits that they may get from the study, I would say outweigh… especially if it's a chronic condition that they've got, that they've lived with for a long time, that I think that if they see a glimmer of hope that that's more important than maybe getting payment. If, though, the study was very… sorry, had a number of visits, I think then that would be where the payments would then move for me. (RN4 – ranked in column 5)

From the ranking summary, PTPs ranked 'Will there be any impact on any insurance policies?' as the second least important statement in position 31 and most did not see the relevance of this item for the decision. RNs ranked this in position 17 with some citing reasons for particular cohorts as influencing their placement.

… I don't know, maybe I'm a bit blasé about that as well. That just didn't come into my head at all. Even at the moment I'm thinking… no just wouldn't affect me one little bit… I think even if I was given an information leaflet on the impact on insurance policies I probably wouldn't even read it, to be honest. (PTP7 – ranked in column 9)

And insurance policies, I think that's important because not all of the patients you have will be in their eighties and not having holidays anymore. So insurance is important for the younger ones, maybe in their fifties or younger, looking to go on holiday. (RN3 – ranked in column 5)

The third least important items ranked in position 30 by PTPs was 'Will expenses be reimbursed?' and again referenced their health as taking precedent over expenses, but it may be important dependent on contribution. However, RNs ranked this statement in position 18 based on real examples of patients being out of pocket and this impacting on recruitment.

That's less important for me, mostly because I wouldn't perceive much in the way of expenses for myself for anything, because I live near the city centre and walk most places… I wouldn't have thought – unless the study happened to be in another city or anything like that – that I would have far to go. (PTP10 – ranked in column 8)

But they [patients] are thinking, and I know the study I'm involved at the moment is involving extra visits for the patients, and I'm expecting that to be a bit of a hurdle if there's not a budget for these extra visits

and parking outside the hospital and things like that. (RN7 – ranked in column 5)

RNs ranked 'Who has approved the study?' as the least important statement in position 32 and in comparison PTPs ranked this statement in position 17. Collectively, RNs seemed to think this was important information for professionals but not for PTPs yet the PTP group placed this higher suggesting it is of value.

… whenever I have been consenting somebody and said where the approvals are from or anything, there's not really any interest at all. (RN1 – ranked in column 9)

I know there's a whole process involved for these things so I wouldn't want to see and I wouldn't really need to know. I would assume it had been properly approved. (PTP1 – ranked in column 5)

The second least important items ranked by RNs in position 31 was 'How have patients and the public been involved in the design of the study?' with PTPs ranking this items at position 26. Both groups recognised the importance of the contribution of patients and the public (although it was not clear if the PTP group fully understood what this item meant) but thought other aspects were more important.

… I don't think patients think about that… I don't think it's of any relevance to them… its obviously important because for a study to work then it has to be in research for a reason and if you have patients involved in the design of it then compliance rates are going to be better. (RN2 – ranked in column 9)

Yeah, I'd be interested in knowing that but I don't think I would immediately want to know how the study had been put together. (PTP9 – ranked in column 7)

The RNs ranked 'Has the scientific quality of study been checked?' as the third least important statement in position 30 largely because in their experience this is not raised as a concern by patients. Interestingly, PTPs ranked this in position 11 stating that these quality checks on research were important. These items had one of the largest variations in ranking between the groups and the largest difference between the groups across the top and bottom three (difference of 19 ranked position (median score difference of 3.5 (PTP = 4.5 vs RN = 8)).

Never had any questions about that. I have had patients or relatives who are well educated, they would want to know the purpose of the study but they would not… They don't want to know overall how many people you require, its more about whether we have any experience doing this thing. (RN5 – ranked in column 9)

I think that would be very important to know. I know there's all sorts of rules about what's a good sample size and things like that, you know, so I would like

to be able to access that information. It wouldn't be as important, I think, as the other things I've ranked highly, but it would be more important. (PTP20 – ranked in column 5)

## Items exhibiting variability on rank order between groups

Figure 1 illustrates the differences between stakeholder groups with regard to median ranking values of informational items ranging from most to least importance. As stated previously, items with a median difference greater than or equal to 2 rank points were considered to have significant variability between the individual groups. Table 4 lists each of the items that exhibited variability in median rank order between the stakeholder groups. Overall, 10 of the 32 items exhibited variability (predefined at ≥2 median scores difference) between the two stakeholder groups on rank order scores. The item with the largest median score rank ordered difference between the PTP and RN group was 'Has the scientific quality of study been checked?' As mentioned previously, there was a 3.5 median score difference between the groups (PTP=4.5 vs RN=8) with PTP ranking it at number 11 and RNs at position 32. One RN provided the following feedback on the exercise, which may provide some explanation as to why differences between the two groups were evident.

What I probably found hard is putting myself maybe say in the patients' shoes, because you can think of it from, you know, very much like, you know, your role as from a nursing perspective, so yeah, always thinking about the patient. (RN4)

## Missing information

On completion of the Q-sort interview, the PTP group were asked whether they felt any information items were missing from the card-sort set. The general consensus was that no additional information items were required, although three participants made suggestions as to additional information they might like to see in a PIL, namely, contact with other patients taking part in the trial; childcare arrangements; and side-by-side comparison between standard care and trial interventions.

### Contact with other patients taking part in the trial

It would be more likely, I think, in some ways, that I would like to have contact because I would…. You know, I think I would appreciate sharing experiences, and I don't know… just thinking about it that might be something that would be useful for the study as well. (PTP10)

### Childcare arrangements

So a logistical question I think is something that I would probably think… it would make me more positive towards something if it said there are facilities for childcare here or there's a crèche or something like that, then it would make me think, 'Oh, well, I can definitely do that then'. (PTP2)

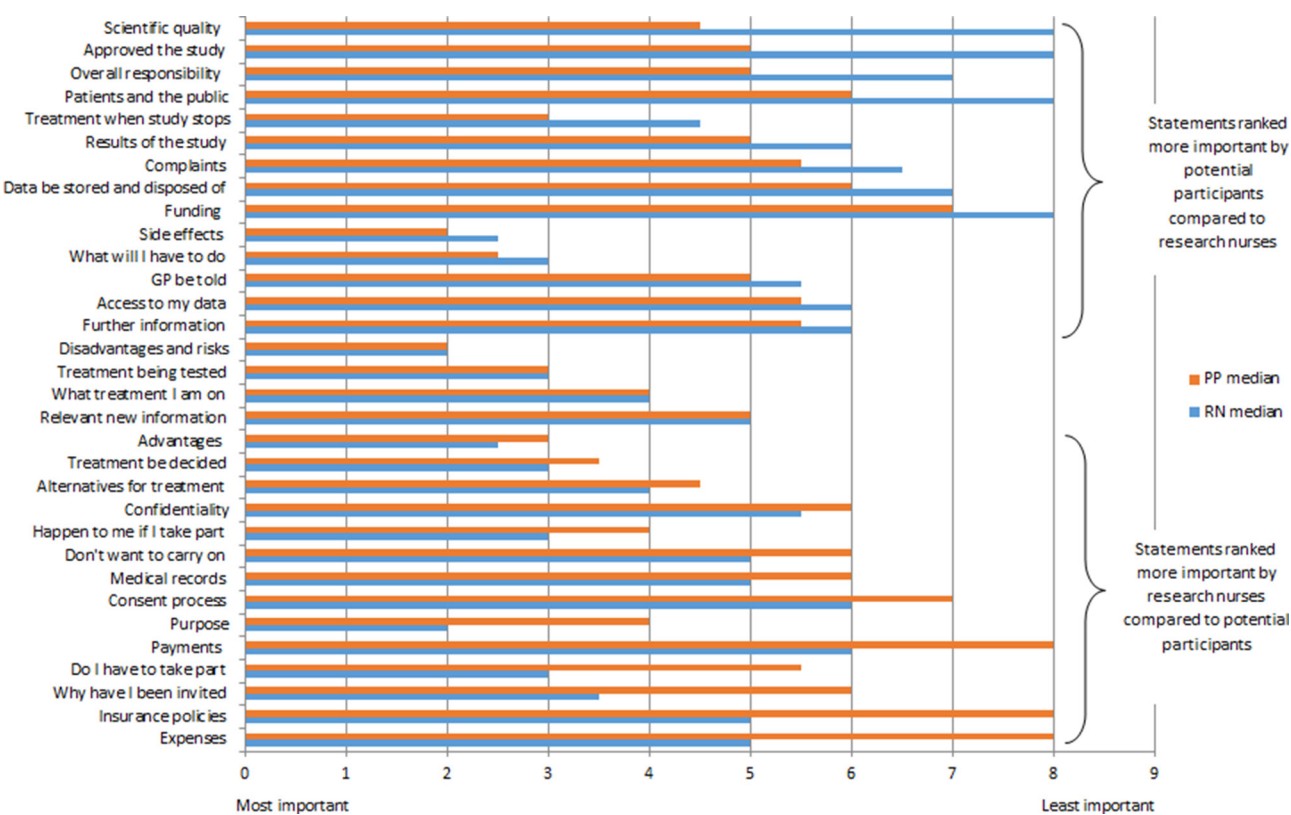

**Figure 1** Median importance scores: comparisons between potential trial participants and research nurses. GP, general practitioner; RN, research nurse.

## Side-by-side comparison between standard care and trial interventions

Maybe exactly what it would entail weighed up against… you know, showing the two side-by-side. This will entail having to come to hospital every week to get bloods, whereas normally you would never have to go and get… how time consuming it would be would probably be quite an important one. (PTP7)

## Minimum information requirement for decision making

On completion of the Q-sort, we asked each of the PTP group if they could indicate at which point they felt they would have enough information to make a decision about taking part in the hypothetical RCT. The median number of cards required by the PTP group to make a decision was 14 with a range of 5–32. For the majority of the PTP group (60% of PTPs), a decision would be made that they

**Table 4** Items exhibiting significant variability on median rank order between stakeholder groups

| | Statement | Median difference | Median score | | Item rank position | |
|---|---|---|---|---|---|---|
| | | | PTP | RN | PTP | RN |
| 1 | Has the scientific quality of study been checked? | 3.5 | 4.5 | 8 | 11 | 32 |
| 2 | Will expenses be reimbursed? | 3 | 8 | 5 | 30 | 18 |
| 3 | Will there be any impact on any insurance policies? | 3 | 8 | 5 | 31 | 17 |
| 4 | Who has approved the study? | 3 | 5 | 8 | 17 | 32 |
| 5 | Why have I been invited? | 2.5 | 6 | 3.5 | 22 | 10 |
| 6 | Do I have to take part? | 2.5 | 5.5 | 3 | 18 | 7 |
| 7 | What is the purpose of this study? | 2 | 4 | 2 | 9 | 2 |
| 8 | Will I receive any payments for taking part? | 2 | 8 | 6 | 32 | 23 |
| 9 | How have patients and the public been involved in the design of the study? | 2 | 6 | 8 | 26 | 31 |
| 10 | Who has overall responsibility for the study? | 2 | 5 | 7 | 16 | 27 |

PTP, potential trial participant; RN, research nurse.

had enough information using between 8 and 15 cards (25%–47% of the 32 statements).

## Interpretation of context

An additional finding from the 'think aloud' interview data relates to participants interpretation of the specific context of the phase III trial described in the vignette. Although no reference to specific interventions was given apart from 'treatment', the majority of participants interpreted the setting to be a drug trial. Examples of this belief were evidenced across both groups.

> My reason is that I just think if you were going to take something that was… if it was going to be taking medication or if it was going to be some other sort of new treatment, it would be important to know as much as you could about what possibly might go wrong with it. (PTP20)

> … So if people getting drug A are clinically much better than the people getting drug B and that's evident quite early on when people would be expected to stop and move on to… (RN2)

## DISCUSSION
### Principal findings

We believe this study to be one of the first to provide evidence in relation to how important PTPs and RNs perceive the informational items prescribed in the regulatory guidance to be with regard to making an informed choice about RCT participation. Our study used a novel methodology in this context (trials methodology) to obtain rankings of informational items for PILs from different trial stakeholder groups, namely PTPs and RNs. Previous research evidencing the relative importance of items included in trial PILs across different stakeholder groups is limited. Existing research on trial PILs has largely assumed the regulatory guidance reflects what potential participants actually want to know and has focused on areas such as structure, content or mode of delivery.[8–10] Our study shows that more work is required to first define *what* information PTPs need (and/or want) to support an informed choice about participation.

Several of the statements identified as being most important relate to information about consequences of participation, namely disadvantages or advantages. Our results are, perhaps, not surprising given various decision-making theories and frameworks suggest that weighing up the pros and cons of a situation is a key component of decision making.[15] In addition, several reports in the literature from qualitative studies that have explored participants' reasons for participation (or not) in RCTs cite potential advantages or disadvantages of the trial as being influential.[16 17] However, it may be important to further consider the context of the trial with regard to relative importance of items. The use of the vignette revealed that although not specified, participants in our study believed the trial to be a drug trial,

which may have influenced how they rated the relative importance of items.

Our results highlight that stakeholder groups were more similar when considering the most important items and that much more variability was exhibited between the groups with regard to the statements considered to be least important. Similar work exploring the importance of informational items included in a decision support intervention for trial participation also identified differences between stakeholder groups on key items.[18] In particular, items describing the advantages or disadvantages of non-participation (eg, forgoing access to trial intervention) in a trial showed more variation than others.[18] An additional study has also evidenced variability among stakeholder groups with regard to content and mode of delivery of information provided to participants to support decisions about trial participation.[19] The differences between stakeholders in perceived importance of information for trial participation decisions is of concern given much of the decision about participation is supported through conversations, which may or may not talk to a PTP's main concerns, depending on who leads that conversation. The coverage of trial topics depending on who leads the conversation has been observed in recruitment consultations for a prostate cancer trial and had implications for recruitment and acceptance of allocation.[20] It is also possible that in practice some RNs adapt their conversation to be responsive to the needs of individual patients and their concerns and preferences for information. Therefore, further research to unpack why differences between stakeholder groups exist and efforts to reduce these differences are important.

The majority of potential participants in our study revealed they would have made a decision about trial participation based on the information items they placed within the first 3–4 most important columns (around 8–15 cards out of 32 and equal to around 47% of the information specified in the HRA guidance). This suggests that all of the information that is included in a PIL may not be necessary for potential participants to make a decision about taking part in the trial. In further support of this, a study that explored the preferred length of the participant information sheet for research showed that 77% of participants chose to access only the first level of information (less than that which may be contained on a standard PIL) before making a decision about participation.[21] In terms of the content of the minimum information set that potential participants deemed sufficient for decision making, our study showed they focused on statements related to the interventions (and any associated consequences) rather than the formalities of the research. These findings are similar to Sand *et al*,[22] who showed that the statements participants valued most were largely related to the study treatment and study-related activities rather than information on storage of data. Whether these key decision statements should be ordered such that they are represented first in PILs requires further research.

As mentioned previously, a systematic review identified little evidence of what information potential participants want to know when making a decision about research participation. Evidence could only be identified, from included studies, for less than half of the items the HRA suggest should be consideration for inclusion in PILs for research.[6] While this review focused more broadly on research studies, not just trials, it further illustrates the point that the information provided in PILs falls short of being actually grounded in the informational needs and desires of those for whom it should be designed. This begs the question of who these patient-facing documents are actually written for. Armstrong et al[7] conducted a study to explore the function of PILs in which they concluded 'PILs are the outcome of a process of institutional scripting that is strongly shaped by the accountability demands inherent in the ethical review process'. They go on to suggest that the content and text of a PIL is agreed between the trialist (the author of the PIL) and the Research Ethics Committee (REC).[7] This lack of recognition of the audience of PILs is further evidenced when comparing PILs for RCTs to other information resources shown to support decision making for treatment and screening decisions (so-called decision aids).[23] PILs were shown to lack information deemed necessary to support good quality decision making.[23] Interestingly, in our study, the PTP group raised 'contact with other participants' and a 'side-by-side comparison of trial treatment and standard care' as being missing from the current information set. Both of these items are suggested as components of decision aids and to be useful for PTPs decision making.[23] Perhaps it is time to review the guidance documents available to researchers to ensure that PILs are written specifically with the needs/wishes of the target audience, the PTP, in mind and that the information more supports informed choices about trial participation with less focusing on institutional accountability.

When patients get involved in the design of research studies, they are frequently asked to help to improve the participant information. There is evidence to show that as potential participants, they can help to make the language clearer and easier to understand and not discriminatory or stigmatising.[24] They can also help to present and deliver the information in ways that reflect the needs of participants and are culturally appropriate and sensitive.[25] There is evidence that involving patients can also help to ensure that the content covers some important aspects of what potential participants want to know but not by systematically examining the information prescribed in national guidance as in the study reported here.[26] In this study, both the PTPs and RNs gave a low ranking to the statement about the involvement of patients and the public in the design of the study. This is not surprising because the statement did not give any indication how the involvement might have helped PTPs make an informed decision whether to participate.

Evidence from research on information to support the informed consent process is needed by the trials community. A recent prioritisation exercise to identify the top 10 research priorities for recruitment in trials identified three priorities in the top 10 that could consider aspects of information provision in their scope.[27] Specifically: priority 2. What information should trialists communicate to members of the public who are being invited to take part in a randomised trial in order to improve recruitment to the trial?; priority 4. What are the best approaches for designing and delivering information to members of the public who are invited to take part in a randomised trial?; and priority 9. What are the best approaches to optimise the informed consent process when recruiting participants to randomised trials?[27] This prioritisation (by a range of stakeholders including patients) of multiple questions around information to support the informed consent process to trials further highlights the need for additional research to identify models of best practice.

### Strengths and limitations

The sample included in this work is relatively small (n=20) and limited by geographic location. Identifying PTPs through the SHARE database was a straightforward, cost-effective and time-saving method; however, it is worth giving consideration to the type of people who have signed up to this database. Those who sign up to the SHARE register are likely to have an interest in research, perhaps making the sample somewhat dissimilar from the general public. The type of information these participants value (or do not value) may differ given their existing experience of research or a more general awareness of research participation. While we have no reason to believe the locality would influence the results, it would be important to extend both the sample size, geographic spread and representation from other stakeholder groups.

Although the vignette was worded slightly differently for each stakeholder group, it was used to try and ensure that the study was interpreted in the same way for all participants. PTPs appeared to have no problems with the vignette as they were being asked to think about a decision from their own point of view. For the RN group, we were asking them to think about what potential participants thought, and this proved more challenging for the RNs. As such, the vignettes between the two groups were slightly different, most notably in the RN group through the use of phrasing around comparing treatments, which was lacking from the PP group. Therefore, it must also be considered that this difference could have influenced preferences for information. Although the vignette talked about treatments—treatment 'a' and treatment 'b'—for a chronic condition, many participants interpreted this as two drug treatments. It is worth considering the possibility that this may have had an impact on how the statements were ranked. For example, information relating to side effects, and risks and disadvantages may be deemed more pertinent for people considering participation in a drug

trial (especially if it were a new product) compared with a trial of non-drug interventions. Further exploration of different aspects of trial design (including different interventions) and how this influences preferences for information is needed. Indeed, the purposive exploration of a range of vignettes that describe different contextual aspects of the trial (eg, uncertainty surrounding each intervention and the risk/benefit profiles for each) would be important to further consider whether context plays a role. Another potential limitation with regard to interpretation relates to the Q-sort statements. Although prompts were developed if participants struggled with interpretation, the statements for the Q-Sort were all quite short and therefore their meaning was open to a certain amount of interpretation. The meaning of each statement and how clear it is may have had a bearing on what the participants understand by it and how important they think it is.

A significant strength of this study was the use of the Q-methodology providing both qualitative and quantitative data to investigate how important different stakeholder groups perceived the informational items to be. The use of Q-methodology in trials methodology research is not common, but the data it produces yield novel insights not easily produced by other methods.[28]

## CONCLUSION

In conclusion, this study has provided a unique insight into how and why different trial stakeholder groups rank informational items contained within PILs for RCTs. This study has shown that both PTPs and RNs ranked similar statements as being most important, yet clear differences exist in the ranking of the least important statements. These results have implications for researchers developing PILs for RCTs. Patient information leaflets are directed at PTPs and should therefore, by default, include information that PTPs want and need to make an informed choice about participation in a trial. Additional efforts to work in parallel with PTPs to identify the information considered critical to support informed choices about trial participation is needed.

**Acknowledgements** The authors would like to thank the stakeholders who participated in the study for their time.

**Contributors** KG was responsible for conceiving the study. KI, SC, MKC and KG designed the study. KI conducted the data collection and statistical analysis. KG and KI conducted the qualitative analysis. KG and KI led the writing of the manuscript. SC, MKC and JE contributed to further drafts of the manuscript. All authors read and approved the final manuscript.

**Funding** This work was supported by personal fellowship award (to KG) from the Medical Research Council Strategic Skills Methodology Fellowship (MRC MR/L01193X/1). KI and SC were supported by awards from the National Institute for Health Research Health Technology Assessment Programme (HTA ref 14/192/71, HTA ref 11/58/15). The Health Services Research Unit is supported by a core grant from the Chief Scientist Office of the Scottish Government Health and Social Care Directorates.

**Competing interests** None declared.

**Patient consent** Not required.

**Ethics approval** The study was approved by NRES Committee London – Bromley (Rec ref: 15/LO/1221) and NHS Grampian Research and Development department (R&D ref: 2015UA013).

**Provenance and peer review** Not commissioned; externally peer reviewed.

**Data sharing statement** No database available.

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
