## [Reviewer comments · BMJ Open]

ARTICLE DETAILS

TITLE (PROVISIONAL)	Relative importance of informational items in Participant Information Leaflets for trials: a Q-Methodology approach.
AUTHORS	Innes, Karen; Cotton, Seonaidh; Campbell, Marion; Elliott, Jim; Gillies, Katie

VERSION 1 – REVIEW

REVIEWER	Peter Knapp University of York & the Hull York Medical School, UK
REVIEW RETURNED	17-May-2018

GENERAL COMMENTS	This is an interesting and potentially very useful manuscript. It presents novel and well-conducted research. The reporting is clear. I have a few comments: 1. In the Background section (page 5) there is discussion of the role of printed and spoken information. It is worth mentioning that consent to research, including to trials, sometimes happens without conversation between the researchers and potential participants (ie in postal recruitment). In that situation the printed information has even more importance.2. Similarly the printed information potentially has another and ongoing role, as a record of what the participant has agreed to. This is mentioned but perhaps needs strengthening.3. The rationale for the limited Q method analysis used, seems reasonable (page 6, lines 35-36).4. The vignettes were very brief. There is a good defence of that decision, in not wanting to lead respondents or overwhelm them with information. However, I am concerned that the lack of content may have influenced respondents' judgements: for example, there is nothing on the uncertainty about which of the 2 treatments is a better option (ie clinical equipoise). Anyone involved in trials would know this would be pre-requisite for a trial but perhaps this isn't obvious to respondents? Similarly, what is the risk of harm of the two treatments? That's important contextual information that could well colour the judgements of the Q study participants.5. One further point is the lack of equivalence of the vignettes given to nurses and participants. While I understand that the vignettes cannot be identical, as they have to place the reader into a situation that allows them to make personal judgements, the content here is different in an important way: the RN vignette refers to 'comparing treatment A and treatment B'. That important information is missing from the patient vignette. In the patient vignette it might be thought
--

	that the two treatments are options or preferences, and not treatments about which there is unknown evidence of (relative) effectiveness, and which need to be compared in research. This may have affected preferences for information expressed in the Q method task. 6. (top of P21) The difference in preferences expressed by RNs and participants is a concern, but would RNs in practice not adapt their conversation to the *individual* patient and the patient's expressed or perceived concerns and needs? 7. (P23) the possibility is raised that the recruitment of participants from SHARE may have influenced data obtained. I do think this is possible and I would have wanted to hear more about the possible consequences: what effects? what trials have they done before? would their positive attitudes to research colour their judgements of desired information (or information not wanted)? 8. Similarly (p23) what effects on the data might have resulted from the respondents perceiving this as a drug trial? Two smaller points:  1. On p7 lines 45-47, the terms 'most important' and 'least important' are used but the terms on the grid are actually 'more important' and 'less important'. 2. I thought the sentence 'This includes...' (lines 13-15, page 5) would benefit from re-writing (three uses of participant / participation!)
--	---

REVIEWER	Wendy Wood University of Southampton, UK
REVIEW RETURNED	24-May-2018

GENERAL COMMENTS	This is an interesting paper showing important insights into how potential participants prioritise the information needed to decide whether to participate in a research study. I wasn't entirely convinced of the value of the input from research nurses. I am not familiar with the methodology used but it was well described and seemed appropriate. The sample size for Q methodology was stated to be a minimum of 2:1 ratio of statements to participants. This study used 32 statements and had two separate groups of participants so I would have expected a minimum of 16 participants in each group. No information is given as to how the participants were selected from the groups of interested individuals in each group - ie 17 interested PTPs but only 10 used. Were they purposively sampled? The authors have identified the limitations of the study and the further research needed in this area. I look forward to seeing the results of future studies from the group. typos Page 2 line 21 - "1o potential trial participants" - replace 'o' with '0' Page 24 line 12 - "...both qualitative and qualitative..." - I assume one should be 'quantitive'
--

VERSION 1 – AUTHOR RESPONSE

	Reviewer #1
1	This is an interesting and potentially very useful manuscript. It presents novel and well-conducted research. The reporting is clear. I have a few comments: In the Background section (page 5) there is discussion of the role of printed and spoken information. It is worth mentioning that consent to research, including to trials, sometimes happens without conversation between the researchers and potential participants (ie in postal recruitment). In that situation the printed information has even more importance. RESPONSE: We thank the reviewer for their kind response. We agree that acknowledging in those trials where recruitment may not involve in-person contact the importance of written information in trials may be greater. CHANGE: Please see sentence included on page 4 lines 78-80. However the recruitment and consent process for some trials is such that a conversation between a researcher and potential participants is less likely (e.g. postal or online recruitment) and here the written information may have more influence.
2	Similarly the printed information potentially has another and ongoing role, as a record of what the participant has agreed to. This is mentioned but perhaps needs strengthening. RESPONSE: We agree that written printed information can also play a supportive role for trial participants as a record of what they agreed to; however, this is a very different function to that of the role of the printed information in the trial decision-making process. Given the focus of the study is on the role of the information in the decision to take part, we would prefer not to amplify this different role further than it is currently. CHANGE: None
3	The rationale for the limited Q method analysis used, seems reasonable (page 6, lines 35-36). RESPONSE: We thank the reviewer for this comment. CHANGE: No response required.
4	The vignettes were very brief. There is a good defence of that decision, in not wanting to lead respondents or overwhelm them with information. However, I am concerned that the lack of content may have influenced respondents' judgements: for example, there is nothing

	on the uncertainty about which of the 2 treatments is a better option (ie clinical equipoise). Anyone involved in trials would know this would be pre-requisite for a trial but perhaps this isn't obvious to respondents? Similarly, what is the risk of harm of the two treatments? That's important contextual information that could well colour the judgements of the Q study participants. RESPONSE: We agree that contextual information could influence responses. We have added an acknowledgement of this point in the discussion CHANGE: Edits for this section have been combined with the edits in response to point 8 below – see revised text under point 8 below.
5	One further point is the lack of equivalence of the vignettes given to nurses and participants. While I understand that the vignettes cannot be identical, as they have to place the reader into a situation that allows them to make personal judgements, the content here is different in an important way: the RN vignette refers to 'comparing treatment A and treatment B'. That important information is missing from the patient vignette. In the patient vignette it might be thought that the two treatments are options or preferences, and not treatments about which there is unknown evidence of (relative) effectiveness, and which need to be compared in research. This may have affected preferences for information expressed in the Q method task. RESPONSE: We agree with the reviewer that the omission of the specific phrase 'comparing treatment A and B' from the potential participant vignette may have led to some of the differences perceived between the groups with regard to preferences for information. We have amended the text to reflect this. CHANGE: See inclusion of 2 sentences on Page 23 lines 687-690. As such the vignettes between the two groups were slightly different, most notably in the RN group through the use of phrasing around comparing treatments which was lacking from the PP group. Therefore it must also be considered that this difference could have influenced preferences for information.
6	(top of P21) The difference in preferences expressed by RNs and participants is a concern, but would RNs in practice not adapt their conversation to the *individual* patient and the patient's expressed or perceived concerns and needs? RESPONSE: We agree that RNs may adapt their practice with regard to the information desires of the patient with whom they are consulting with. We have now included text within the manuscript to acknowledge this point. CHANGE: See Page 21 lines 596-598. It is also possible that in practice some RNs adapt their conversation to be responsive to the needs of individual patients and their concerns and preferences for information
7	(P23) the possibility is raised that the recruitment of participants from SHARE may have influenced data obtained. I do think this is possible and I would have wanted to hear more about the possible consequences: what effects? what trials have they done before? would their positive attitudes to research colour their judgements of desired information (or information not wanted)?

	RESPONSE: We have included an additional sentence in this section to highlight that existing experiences of research or a more general awareness of research may influence preferences. Unfortunately we do not know the type of research studies (e.g. trial versus cohort etc) that the 3 potential participants had been involved in, just they had participated. CHANGE: See inclusion of sentence on Page 23 lines 676-677 The type of information these participants value (or do not value) may differ given their existing experience of research or a more general awareness of research participation.
8	Similarly (p23) what effects on the data might have resulted from the respondents perceiving this as a drug trial? RESPONSE: This response also links back to comment #4 relating to context, and further supports the justification for trying to keep the vignette's broad in scope. However, we agree with the reviewer that it is important to highlight to the reader what consequences the trial being perceived as a drug trial may pose for preferences of information. CHANGE: We have included three sentences on Page 24 lines 693-700 to cover this. For example, information relating to side effects, and risks and disadvantages may be deemed more pertinent for people considering participation in a drug trial (especially if it were a new product) compared to a trial of non-drug interventions. Further exploration of different aspects of trial design (including different interventions) and how this influences preferences for information is needed. Indeed, the purposive exploration of a range of vignettes that describe different contextual aspects of the trial (e.g. uncertainty surrounding each intervention, the risk/benefit profiles for each, etc) would be important to further consider whether context plays a role.
9	Two smaller points: On p7 lines 45-47, the terms 'most important' and 'least important' are used but the terms on the grid are actually 'more important' and 'less important'. RESPONSE: We have edited this sentence to amend the discrepancy highlighted above. CHANGE: See page 7 lines 183-184.
	I thought the sentence 'This includes...' (lines 13-15, page 5) would benefit from re-writing (three uses of participant / participation!) RESPONSE: This sentence has now been edited. CHANGE: See page 5 lines 101-102 This includes those participants who have consented and been recruited to trials and those who are considering participating in trials.

	Reviewer #2
1	This is an interesting paper showing important insights into how potential participants prioritise the information needed to decide whether to participate in a research study. I wasn't entirely convinced of the value of the input from research nurses. I am not familiar with the methodology used but it was well described and seemed appropriate. The sample size for Q methodology was stated to be a minimum of 2:1 ratio of statements to participants. This study used 32 statements and had two separate groups of participants so I would have expected a minimum of 16 participants in each group. RESPONSE: We thank the reviewer for their considered comments. The ratio of statements to participants does not distinguish between different stakeholder groups. We have amended the manuscript to make this clearer. CHANGE: See page 8 lines 191-193 Typically, Q methodology uses relatively small samples of participants and the literature suggests that a 2:1 ratio of statements to participants (irrespective of stakeholder group) is favoured as a minimum. For example, a study with 40 statements would have 20 participants in total as a minimum
2	No information is given as to how the participants were selected from the groups of interested individuals in each group - ie 17 interested PTPs but only 10 used. Were they purposively sampled? RESPONSE: We thank the review for highlighting this lack of clarity regarding participant selection in the methods section. We have now amended the text to account for this. CHANGE: See page 8 lines 206-211. The details of 17 potentially interested participants were provided to the research team by SHARE. All 17 potential participants were contacted by the researcher by telephone to arrange a convenient time for a Q-sort interview. Following this conversation, ten participants expressed interest (seven declined further information) and were sent postal confirmation of the appointment time and a PIL for the Q-methodology study (available from the researchers on request).
3	The authors have identified the limitations of the study and the further research needed in this area. I look forward to seeing the results of future studies from the group. RESPONSE: Thank you. CHANGE: No response required.

4	typos Page 2 line 21 - "1o potential trial participants" - replace 'o' with '0' Page 24 line 12 - "...both qualitative and qualitative..." - I assume one should be 'quantitive' RESPONSE: Apologies for the typos. These have now been corrected. CHANGE: Page 2 line 28 - 10 potential trial participants Page 24 line 708 - both qualitative and quantitative.

VERSION 2 – REVIEW

REVIEWER	Peter Knapp University of York & the Hull York Medical School, UK.
REVIEW RETURNED	11-Jul-2018

GENERAL COMMENTS	The authors have responded appropriately to all peer review comments
--

REVIEWER	Wendy Wood University of Portsmouth, UK
REVIEW RETURNED	24-Jul-2018

GENERAL COMMENTS	I am happy with the responses to my previous comments
---